# Dopamine Transporter, PhosphoSerine129 α-Synuclein and α-Synuclein Levels in Aged LRRK2 G2019S Knock-In and Knock-Out Mice

**DOI:** 10.3390/biomedicines10040881

**Published:** 2022-04-12

**Authors:** Chiara Domenicale, Daniela Mercatelli, Federica Albanese, Salvatore Novello, Fabrizio Vincenzi, Katia Varani, Michele Morari

**Affiliations:** 1Department of Neuroscience and Rehabilitation, University of Ferrara, 44121 Ferrara, Italy; chiara.domenicale@unife.it (C.D.); daniela.mercatelli@unife.it (D.M.); federica.albanese@unife.it (F.A.); salvatore.novello@epfl.ch (S.N.); 2Technopole of Ferrara, LTTA Laboratory for Advanced Therapies, 44121 Ferrara, Italy; 3Laboratory of Molecular and Chemical Biology of Neurodegeneration, Brain Mind Institute, Ecole Polytechnique Fédérale de Lausanne (EPFL), CH-1015 Lausanne, Switzerland; 4Department of Translational Medicine, University of Ferrara, 44121 Ferrara, Italy; fabrizio.vincenzi@unife.it (F.V.); katia.varani@unife.it (K.V.)

**Keywords:** DAT, G2019S knock-in, [^3^H]-DA uptake, D1994S kinase-dead, LRRK2, MLi-2, pSer129 α-synuclein, Parkinson’s disease, VMAT2

## Abstract

The G2019S mutation in leucine rich-repeat kinase 2 (LRRK2) is a major cause of familial Parkinson’s disease. We previously reported that G2019S knock-in mice manifest dopamine transporter dysfunction and phosphoSerine129 α-synuclein (pSer129 α-syn) immunoreactivity elevation at 12 months of age, which might represent pathological events leading to neuronal degeneration. Here, the time-dependence of these changes was monitored in the striatum of 6, 9, 12, 18 and 23-month-old G2019S KI mice and wild-type controls using DA uptake assay, Western analysis and immunohistochemistry. Western analysis showed elevation of membrane dopamine transporter (DAT) levels at 9 and 12 months of age, along with a reduction of vesicular monoamine transporter 2 (VMAT2) levels at 12 months. DAT uptake was abnormally elevated from 9 to up to 18 months. DAT and VMAT2 level changes were specific to the G2019S mutation since they were not observed in LRRK2 kinase-dead or knock-out mice. Nonetheless, dysfunctional DAT uptake was not normalized by acute pharmacological inhibition of LRRK2 kinase activity with MLi-2. Immunoblot analysis showed elevation of pSer129 α-syn levels in the striatum of 12-month-old G2019S KI mice, which, however, was not confirmed by immunohistochemical analysis. Instead, total α-syn immunoreactivity was found elevated in the striatum of 23-month-old LRRK2 knock-out mice. These data indicate mild changes in DA transporters and α-syn metabolism in the striatum of 12-month-old G2019S KI mice whose pathological relevance remains to be established.

## 1. Introduction

The c.6055G>A transition in the leucine-rich repeat kinase 2 (LRRK2) gene results in the G2019S substitution in the LRRK2 protein, which is associated with familial, autosomal dominant forms of Parkinson’s Disease (PD) and represents a risk factor for idiopathic PD [1,2]. LRRK2 is a multidomain protein encompassing a kinase and a GTPase domain surrounded by protein–protein interaction domains [3,4]. The G2019S mutation causes a 2–3-fold increase in kinase activity which is instrumental to LRRK2 neurotoxicity in vitro [5,6,7] and in vivo [8,9,10]. LRRK2 G2019S-associated PD and idiopathic PD share clinical and neuropathological features. Indeed, they are characterized by hypokinetic symptoms, nigrostriatal dopamine (DA) neuron degeneration and Lewy body pathology [1,11], although the neuropathology in the LRRK2 brains appears more heterogeneous than in idiopathic PD brains [1]. Several rodent models have been developed to recapitulate clinical and neuropathological features of PD [12], although, in most cases, the outcome has been disappointing. In fact, G2019S knock-in (KI) mice [13,14], human G2019S (hG2019S) LRRK2 BAC overexpressing mice [15,16,17], and rats [18,19,20] do not show nigrostriatal degeneration. Only in 12–21-month-old mice overexpressing hG2019S under the PDGF promoter was a 18–50% neuronal loss in SNc consistently reported [21,22]. Nonetheless, abnormalities in DA neuron morphology, DA homeostasis, or transmission have been observed in G2019S LRRK2 KI and transgenic rodents, which have been viewed as early signs of DA neuron demise. Consistent with this view, G2019S KI mice were found to be more prone to develop nigrostriatal degeneration after exposure to Parkinsonian triggers, such as mitochondrial toxins rotenone or MPTP [23,24], genetic or virally-mediated overexpression of mutant α-synuclein (α-syn) [25,26], or injection of preformed fibrils (PFFs) of α-syn [27]. In this context, we reported that G2019S KI mice showed age-dependent elevation of striatal levels and activity of the membrane DA transporter (DAT) along with reduction of striatal levels of the vesicular monoamine transporter type 2 (VMAT2) [14]. It has long been recognized that abnormal synaptic handling of DA, e.g., resulting from DA transporter dysfunction, can result in accumulation of intrasynaptic DA, build-up of DA metabolism byproducts and toxicity of DA terminals [28,29]. DA accumulation resulting from DAT upregulation [30] and/or VMAT2 deletion [31,32] is associated with DA neuron loss, whereas VMAT2 overexpression is neuroprotective [32,33]. Reduced expression of VMAT2 via RNAi in rats [32] leads to the appearance of a Parkinsonian phenotype and focal loss of tyrosine hydroxylase (TH) positive terminals in striatum, and is associated with increased phosphorylation of α-syn at Serine 129 (pSer129 α-syn) in SNc, which can be viewed as an early index of neuronal toxicity [34]. Interestingly, we also reported that DAT and VMAT2 changes in the striatum of 12-month-old G2019S KI mice were accompanied by an increase of pSer129 αsyn inclusions in striatum and SNc [14]. In this follow-up study, to correlate changes of DA transporters with α-syn inclusions, striatal DAT and VMAT2 levels and activity were monitored along with endogenous pSer129 α-syn immunoreactivity in G2019S KI mice of 6, 9, 18 and 23 months and age-matched WT controls. To further investigate the role of LRRK2 kinase activity in DAT changes, mice bearing a kinase-dead mutation (KD mice) and mice with constitutive deletion of LRRK2 (KO mice) were used. Moreover, the LRRK2 inhibitor MLi-2 was acutely administered to reveal whether kinase activity sustains the changes of DAT activity.

## 2. Materials and Methods

### 2.1. Animals

Male homozygous LRRK2 G2019S KI, LRRK2 KO and D1994S KI mice (kinase-dead, KD mice), backcrossed on a C57Bl/6J background, were used in comparison with WT control mice [14,35,36,37]. Colonies were raised at the animal facility (LARP) of the University of Ferrara, under regular lighting conditions (12 h light/dark cycle) with food and water ad libitum. Founders were obtained from Mayo Clinic (Jacksonville, FL, USA) (LRRK2 KO mice) [38] and from Novartis Institutes for BioMedical research (Novartis Pharma AG, Basel, Switzerland) (G2019S KI and KD mice) [39]. Experimental procedures were approved by the Ethical Committee of the University of Ferrara and the Italian Ministry of Health (licenses n. 102/2017-PR and 134/2017-PR). 

### 2.2. Western Blot Analysis

Mice were anesthetized and euthanized by cervical dislocation. Striata were processed as previously described [37] [40] Samples were tested for the following primary antibodies: rabbit anti-DAT (D6944, 1:1000; Sigma Aldrich, Saint Louis, Missouri, USA), rabbit anti-VMAT2 (V9014, 1:300; Sigma Aldrich, Saint Louis, MO, USA), rabbit anti-pSer129 α-syn (ab51253, 1:2000; Abcam, Cambridge, UK), rabbit anti-α-syn (ab212184, 1:1000; Abcam, Cambridge, UK), and anti-rabbit GAPDH (PA1-988, 1:1000; Thermo Scientific, Carlsbad, CA, USA). After incubation with secondary antibody (goat anti-rabbit IgG HRP-conjugate 12–348, 1:4000; Merck Millipore, Darmstadt, Germany), immunoreactive proteins were visualized by enhanced chemiluminescence (ECL) detection kit (PierceTM BCA Protein Assay Kit, Thermo Scientific or ECL+, GE Healthcare) and images acquired and quantified using the ChemiDoc MP System and the ImageLab Software (Bio-Rad, Hercules, CA, USA). Data were analyzed by densitometry, and the optical density of specific target protein bands was normalized to the housekeeper protein level.

### 2.3. DA Uptake Assay

Mice were anesthetized and euthanized by cervical dislocation. Striatal synaptosomes were obtained and DAT activity assay carried out as previously described [14]. Essentially, synaptosomes were incubated for 5 min at 37 °C with 20 nM [^3^H]-DA isotopically diluted with varying concentrations of unlabeled DA to obtain final DA concentrations in the 20–2000 nM range. Non-specific DA uptake was evaluated in the presence of 5 μM GBR-12783 dihydrochloride (Tocris Bioscience, Bristol, UK). The filter-bound radioactivity was counted using a PerkinElmer Tri Carb 2810 TR scintillation counter (PerkinElmer, Waltham, MA, USA). Specific DA uptake was expressed as pmol/mg protein/min [41], and kinetic parameters (Vmax and Km) were determined using Prism 5.0 (Graph-Pad Software Inc., San Diego, CA, USA).

### 2.4. VMAT2 Activity Assay

Whole-brain synaptic vesicles were obtained, and VMAT2 activity assay was carried out as previously described [14]. Synaptic vesicles were incubated for 5 min at 37 °C with 20 nM [^3^H]-DA isotopically diluted with varying concentrations of unlabeled DA. Radioactivity was counted with a PerkinElmer Tri Carb 2810 TR scintillation counter.

### 2.5. Immunohistochemistry

Mice were deeply anesthetized with isoflurane and transcardially perfused with 4% paraformaldehyde in Phosphate Buffer Solution (PBS; 0.1 M, pH 7.4). Brains were removed, transferred to a 30% sucrose solution in PBS for cryoprotection and then stored at −80 °C. Fifty micrometer free-floating sections of striatum (AP from +1.42 to +0.14 from bregma) [42] were rinsed in PBS incubated for 30 min at room temperature with a blocking solution (PBS + BSA 1:50 + Triton X100 0.3%) and then incubated with a rabbit polyclonal antibody raised against total α-syn (ab52168; 1:250 in BSA 1% PBST; Abcam, Cambridge, UK), pSer129 α-syn (ab51253; 1:200 in BSA 1% PBST; Abcam, Cambridge, UK) or DAT (ab184451; 1:300 in BSA 1% PBST; Abcam, Cambridge, UK) overnight at room temperature. Sections were then rinsed and incubated with an anti-rabbit HRP-conjugated secondary antibody (ab6721, 1:500 in BSA 1% PBST; Abcam, Cambridge UK) and revealed by a DAB substrate kit (ab64238, Abcam, Cambridge, UK). Sections were mounted on gelatinized slides, dehydrated and coverslipped for further analysis. Images were taken at 2.5× magnification with a Leica DM6B motorized microscope (Leica Microsystems, Milan, Italy), and optical density was determined offline with ImageJ software.

### 2.6. Drugs

MLi-2 was purchased from Carbosynth Limited (Compton, Berkshire, UK) and was dissolved in 4% DMSO and 30% hydroxyprpyl β-cyclodextrin.

### 2.7. Data Presentation and Statistical Analysis

Data are expressed as mean ± SEM (standard error of the mean) of *n* mice. Statistical analysis was implemented in Graphpad Prism 8.4.3 (GraphPad; LaJolla, CA, USA). The Student *t*-test, two-tailed for unpaired data, was used to compare two groups; in the remaining cases, one-way or two-way ANOVA followed by the Tukey test for multiple comparisons was used. Outliers were identified using the ROUT method implemented in Graphpad Prism 8.4.3 (GraphPad; LaJolla, CA, USA). *p* < 0.05 values were considered statistically significant.

## 3. Results

### 3.1. Time-Dependent DAT Dysfunction in G2019S KI Mice

DAT level and activity were measured in striatal lysates from 6, 9, 12 and 18-month-old G2019S KI and age-matched WT mice (Figure 1). Immunoblot analysis revealed no genotype difference in DAT levels in 6-month-old mice (Figure 1A). Conversely, an increase was observed in 9-month-old (+23%, *t* = 4.36, df = 10, *p* = 0.0015; Figure 1B) and 12-month-old (+20%, *t* = 2.89, df = 10, *p* = 0.0159; Figure 1C) G2019S KI mice compared to WT controls. Such difference, however, disappeared at 18 months (Figure 1D). Whether changes in DAT levels were accompanied by changes in [^3^H]-DA uptake kinetics was investigated next (Figure 2). Affinity (Km) and maximal transport rate (Vmax) were similar between G2019S KI and WT mice at 6 months of age (Km 78.04 ± 3.35 vs. 83.01 ± 5.45 nM, Vmax 24.01 ± 1.82 vs. 23.08 ± 0.92 nM, respectively) (Figure 2A). Conversely, a significant 24% increase of Vmax was found in 9-month-old G2019S KI mice compared to age-matched WT controls (27.54 ± 1.18 vs. 22.25 ± 0.99 pmol/mg prot/min, respectively, *t* = 2.99, df = 6, *p* = 0.0242) which was not accompanied by changes of Km (79.75 ± 8.46 vs. 69.61 ± 8.85 nM, respectively) (Figure 2B). Likewise, 12-month-old G2019S KI mice showed a 62% increased of Vmax compared to age-matched WT controls (32.02 ± 0.71 vs. 19.75 ± 1.22 pmol/mg prot/min, respectively, *t* = 8.26, df = 6, *p* = 0.0020) without changes in Km (69.78 ± 6.67 vs. 67.99 ± 9.15 nM, respectively) (Figure 2C). At 18 months, Vmax was slightly but still significantly higher (+15%) in G2019S KI mice compared to age-matched WT controls (34.70 ± 1.02 vs. 30.27 ± 1.25 pmol/mg prot/min, respectively, *t* = 2.73, df = 6, *p* = 0.0342) whereas Km remained unchanged (75.45 ± 6.48 vs. 76.07 ± 4.29 nM, respectively) (Figure 2D). 

### 3.2. DAT Abnormalities Are Associated with the G2019S Mutation but Independent of Ongoing LRRK2 Kinase Activity

To verify whether the alterations of DAT levels observed in G2019S KI mice were specifically related to the G2019S mutation, striatal DAT levels were measured in a different cohort of 12-month-old G2019S KI mice in comparison with age-matched LRRK2 KO, KD and WT mice (Figure 3A). Immunoblot analysis revealed changes across genotypes (F_3,20_ = 4.02, *p* = 0.0216), and confirmed the elevation of DAT levels in G2019S KI mice (+46%). Conversely, no changes were observed in LRRK2 KO and KD mice compared to WT controls. 

Since immunoblot data suggested that the increase of DAT levels relied on the G2019S mutation, the possibility was tested that the abnormal increase of DAT activity observed in G2019S KI mice was sustained by ongoing LRRK2 kinase activity. [^3^H]-DA uptake kinetics were then analyzed in striatal synaptosomes obtained from 12-month-old G2019S KI and WT mice acutely treated with MLi-2 (10 mg/kg, i.p.) or vehicle (Figure 3B). MLi-2 failed to normalize the elevation of Vmax in G2019S KI mice (genotype effect F_1,12_ = 31.44, *p* = 0.0001; MLi-2 effect F_1,12_ = 0.0008, *p* = 0.99). To confirm effective LRRK2 targeting by MLi-2, striatal pSer1292 LRRK2 levels were monitored in G2019S KI mice as a readout of kinase activity [43]. MLi-2 caused a 75% reduction of pSer1292 LRRK2 levels (t = 8.838 df = 10, *p* < 0.0001; Figure 3C) without affecting total protein levels (Figure 3D), indicating that MLi-2 impaired striatal LRRK2 kinase activity.

To confirm the elevation of DAT signal observed in immunoblot experiments, we performed DAT immunohistochemistry in the striatum of 12-month-old G2019S KI mice. Although DAT levels were slightly higher in G2019S KI and LRRK2 KO mice with respect to controls, ANOVA did not reveal significant changes in DAT density across genotypes (F_2,15_ = 1.12, *p* = 0.35; Appendix A). Immunohistochemistry was repeated in striatal slices of 23-month-old mice (Appendix A). DAT immunoreactivity was unchanged across genotypes (F_2,21_ = 2.707, *p* = 0.089) although the DAT signal trended to a 7% reduction in G2019S KI mice with respect to controls (*p* = 0.09).

### 3.3. VMAT2 Dysfunction in G2019S KI Mice

VMAT2 protein levels were analysed in the striatum of 6, 9, 12 and 18-month-old WT and G2019S KI mice (Figure 4). VMAT2 levels were similar between genotypes in 6-month-old and 9-month-old mice (Figure 4A,B), whereas a 53% decrease was found in 12-month-old G2019S KI mice (*t* = 9.806, df = 10, *p* < 0.0001; Figure 4C) [14]. VMAT2 protein levels were similar between genotypes at 18 months (Figure 4D). To investigate the LRRK2 kinase dependence of the effect observed, VMAT2 levels were measured in G2019S KI mice in comparison with age-matched LRRK2 KD, KO and WT mice. Marked (−55%) reduction of VMAT2 levels was confirmed in the new cohort of G2019S mice, whereas no changes were observed in KD and KO mice (F_3,20_ = 6.487, *p* = 0.0030; Figure 4E). VMAT2 uptake activity was then analysed in a whole-brain synaptic vesicle preparation of 9-month-old and 18-month-old mice (Figure 5). The analysis of [^3^H]-DA uptake kinetics revealed no difference of Km and Vmax between G2019S KI and WT mice at 9 months (Km 352.10 ± 33.80 vs. 366.40 ± 21.00 nM, Vmax 44.31 ± 2.70 vs. 41.93 ± 1.89 nM, respectively) (Figure 5A) or 18 months (Km 379.6 ± 20.0 vs. 359.6 ± 15.5 nM, Vmax 40.70 ± 3.05 vs. 44.78 ± 1.18 nM, respectively) (Figure 5B).

### 3.4. pSer129 α-Syn and Total α-Syn in G2019S KI and LRRK2 KO Mice

Immunoblot analysis was performed in the striatum of 12-month-old WT, G2019S KI and LRRK2 KO mice, to investigate whether changes in DAT levels were associated with changes in total α-syn and pSer129 α-syn levels (Figure 6). Total α-syn levels (normalized to the houskeeping gene GAPDH) were unchanged across genotypes (F_2,23_ = 0.06, *p* = 0.91; Figure 6A), whereas pSer129 levels (Figure 6B) were almost doubled in G2019S KI mice when compared to both WT and LRRK2 KO mice (F_2,23_ = 4.69, *p* = 0.019; Figure 6B). However, only a trend towards an increase was observed when pSer129 α-syn levels were normalized to total α-syn levels since ANOVA yielded a *p*-value just above the limit of significance (F_2,23_ = 2.93, *p* = 0.07, Figure 6C).

α-syn immunohistochemistry was finally performed in striatal slices from 12, 18 and 23-month-old mice (Figure 7, Figure 8 and Figure 9). ANOVA revealed a significant change of pSer129 α-syn immunoreactivity across genotypes (F_2,15_ = 3.916, *p* = 0.042; Figure 7A), and post-hoc analysis revealed a slight increase (~6%) of pSer129 α-syn immunoreactivity in 12-month-old LRRK2 KO mice compared to age-matched WT controls. Instead, no difference was found between G2019S KI and WT mice or G2019S KI and LRRK2 KO mice. Conversely, total α-syn immunoreactivity remained unchanged across genotypes (F_2,15_ = 2.753, *p* = 0.09; Figure 7B). At 18 months, striatal pSer129 α-syn and total α-syn immunoreactivity were similar in G2019S KI and WT mice, although total α-syn trended to increase in G2019S KI mice (+14%, *t* = 2.128, df = 14, *p* = 0.051, Figure 8A,B). At 23 months, pSer129 α-syn staining was similar across genotypes (F_2,26_ = 0.51, *p* = 0.60; Figure 8A), whereas total α-syn levels were significantly different (F_2,26_ = 7.735, *p* = 0.0023; Figure 9B). In particular, LRRK2 KO mice showed a significant ~15% elevation in comparison to WT mice but no statistically significant difference with G2019S KI mice. Interestingly, however, G2019S KI mice showed a ~10% increase compared to WT controls, which, however, was just above the limit of significance (*p* = 0.058).

## 4. Discussion

A temporal analysis of DA transporter levels and activity in relation to α-syn and pSer129 α-syn immunoreactivity was performed in the striatum of G2019S KI mice. DAT dysfunction emerged earlier than VMAT2 dysfunction (as early as age 9 months) and outlasted it. The increase of DAT and the reduction of VMAT2 levels were observed in G2019S KI mice but not mice where LRRK2 was deleted or LRRK2 kinase activity silenced, suggesting that they are related to the mutation-associated increase of LRRK2 kinase activity. Nonetheless, the abnormal DAT activity was independent of ongoing LRRK2 kinase activity since MLi-2 failed to normalize it at a dose reducing pSer1292 levels by 75% [36,44]. The most parsimonious explanation is that abnormal DAT activity ensues from permanent changes associated with chronic enhancement of LRRK2 kinase activity at the dopaminergic synapse. Nonetheless, the possibility that repeated MLi-2 administrations are necessary to normalize DAT activity needs to be tested.

Previous studies in LRRK2 mice revealed that LRRK2 modulates DAT levels and activity [14,20,45] (for a review see [46]). A dampened response of striatal DA release to DAT blockers was detected in R1441R KI mice [47], G2019S KI mice [14] and 18-month-old hG2019S rats overexpressing the transgene in the adulthood [20]. Paradoxically, here we confirm that the response in G2019S KI mice is associated with an increase of DAT protein levels and DAT velocity (Vmax) without changes in affinity for DA [14]. As it has been shown that the density of DAT expressed in plasma membranes inversely correlates with sensitivity to DAT blockers [48,49], we have speculated that the reduced sensitivity to DAT blockers might indeed reflect a greater accumulation of DAT, and possibly DAT uptake activity, at the plasma membrane. Immunohistochemistry failed to replicate the results of Western analysis since DAT immunoreactivity was non-significantly elevated in striatal slices of 12-month-old G2019S KI mice. However, it should be recalled that, despite very sensitive, indirect immunohistochemistry is not quantitative and the signal cannot be unequivocally referred to a specific protein as in Western analysis. Anyway, the lack of changes of DAT immunoreactivity would suggest that DAT changes are mild. Elevated DAT activity has been associated with cell death in vitro [30,50,51,52] and experimental Parkinsonism in vivo [30] due to facilitation of toxin (e.g., MPTP) entry and/or buildup of cytosolic DA resulting in the production of toxic species and oxidative stress [30,50,51,53]. This process might be amplified by the impairment of VMAT2 activity [28] since, consistent with the VMAT2 role in maintaining cytosolic DA concentrations within a physiological subtoxic range, deletion of VMAT2 results in DA neuron loss [31,32,54], whereas VMAT2 overexpression is neuroprotective [31,33]. The divergent nature of the time-courses of DAT and VMAT2 levels in G2019S KI mice indicates that they are not related to changes of striatal DA terminal density. Indeed, striatal TH density does not change in G2019S KI mice compared to WT controls over aging [13,14]. It is also difficult to infer whether changes in DA transporters are linked to aging or to strain differences since across-genotype and not within-genotype longitudinal design of immunoblot analysis was performed. However, DA homeostasis undergoes age-related changes in G2019S KI mice and, specifically, DA transmission appears to be elevated at 3 but not at 12 months of age [45]. Since vesicular DA release appears similar in 12-month-old WT and G2019S KI mice [14,45], the elevation of DAT activity we observed in G2019S KI mice at 12 months might explain the normalization of DA transmission observed at this age [45]. The fact the functional changes in DA uptake outlast changes in protein levels might be due to the higher sensitivity of the functional assay, since the immunoblot analysis cannot discriminate between active and inactive DAT pools. It is unlikely that dysregulation of DA homeostasis ensues in nigral DA cell loss since G2019S KI mice do not show overt nigrostriatal DA neuron or terminal degeneration during their lifespan [13,14]. Nonetheless, they show signs of synaptic dysregulation [14,35,55] and increased susceptibility to Parkinsonian toxins along with aging [40], suggesting that they can model the presymptomatic/premotor phases of the disease. In support of this view, changes in DAT and VMAT2 levels/activity in the striatum of 12-month-old G2019S KI mice were associated with an increased phosphorylation of α-syn at Ser129, as revealed by immunoblot analysis. This can be considered an early sign of neuron demise. In fact, although the role of this posttranslational α-syn modification in neurodegeneration remains to be established [34], presynaptic pools of pSer129 α-syn unrelated to neurodegeneration have been identified [56]. Previous dual immunofluorescence analysis revealed that the increase of striatal pSer129 levels in G2019S KI mice occurs in neurons [37], consistent with the view that G2019S LRRK2 might orchestrate DAT, VMAT2, and α-syn changes within the dopaminergic synapse. The present study cannot prove a causal link between DAT and pSer129 α-syn changes or tell whether dysregulation of DA homeostasis precedes the elevation of Ser129 α-syn phosphorylation or viceversa. Both hypotheses might be proven true. In fact, partial genetic knockdown (~50%) of VMAT2 in rats has been reported to bring about impaired striatal DA transmission, nigro-striatal DA cell loss and increased pSer129 α-syn levels [32]. On the contrary, it is well known that α-syn binds to DAT [52,57,58,59] and that α-syn regulates DAT trafficking [58,60] and surface localization, which is key to the modulation of DA uptake and DA homeostasis. Both positive [52,61,62] and negative [57,59,63] α-syn modulations of DAT function have been reported. The positive association of DAT levels, DAT uptake and pSer129 α-syn levels in aged G2019S KI mice is consistent with a study in SH-SY5Y cells showing that pSer129 α-syn increased Vmax without changing Km of [^3^H]-DA uptake, suggesting that pSer129 α-syn increases DA uptake independent of DAT trafficking [62]. Therefore, although the present data cannot prove a causal link between DAT, VMAT2 and pSer129 α-syn changes, they suggest that these changes might be part of an overall (synaptic) adaptive response associated with the G2019S mutation. pSer129 α-syn immunoreactivity analysis did not confirm immunoblot data, again revealing different sensitivities of the two techniques. The discrepancy with previous immunohistochemical finding in G29019S KI mice [26] might be due to the different method of pSer129 α-syn quantification (area-of-threshold vs. optical density) and the different model used (AAV-GFP-injected vs. naïve G2019S KI mice). Immunohistochemistry, however, revealed an increase in α-syn immunoreactivity in the striatum of 23-month-old LRRK2 KO mice. This is consistent with the elevation of α-syn levels in the kidneys of 15-month-old LRRK2 KO mice [64], although no such increase was observed in the brain of these mice. Conversely, α-syn immunoreactivity was elevated in the striatum of 15-month-old LRRK1 and LRRK2 double KO mice compared to LRRK1 and LRRK2 single KO mice and WT controls [65]. In this respect, the present data would suggest that LRRK1 compensation of LRRK2 functions [65] is not lifelong and might fade at very late ages (i.e., 23 months) leaving LRRK2 function uncompensated.

## 5. Conclusions

We previously reported aberrant DAT and VMAT2 levels and dysfunctional DAT in 12-month-old G2019S KI mice [14]. Here, a careful time-course of DAT and VMAT2 expression/activity and α-syn immunoreactivity in G2019S KI mice up to 23 months of age was carried out. In confirming aberrant DAT and VMAT2 levels and dysfunctional DAT in 12-month-old G2019S KI mice, we show for the first time that DAT changes preceded and outlasted VMAT2 changes, and that DAT dysfunction is not sustained by ongoing LRRK2 kinase activity. Various clinical studies failed in showing changes of DAT imaging in asymptomatic LRRK2 mutant carriers [66,67,68]. Nonetheless, a more recent analysis performed in a significantly larger and homogeneous patient population showed a better PET ligand binding to DAT in LRRK2 G2019S non-symptomatic carriers with respect to sporadic PD patients, which has been interpreted as being due to a slower DAT decline or a reduced DA release [69]. Our data lend support to the hypothesis that better DAT signal is due to an increase in DAT availability, although the possibility that G20919S KI mice also have reduced DA levels cannot be ruled out [13,14]. Further studies are needed to explore the mechanisms by which the G2019S mutation regulates DAT function. Immunoblot analysis in the striatum of 12-month-old G2019S KI mice also confirms the elevation of pSer129 α-syn levels previously reported with immunohistochemistry [14], although a different quantification method of immunostaining can influence the outcome of the analysis. Finally, immunohistochemical analysis extended from 12-month-old G2019S KI and WT mice [14] to 12, 18 and 23-month-old G2019S KI, LRRK2 KO and WT mice reveals for the first time a very late elevation of α-syn levels in LRRK2 KO mice, which confirms the role of LRRK2 in α-syn clearance and, more in general, proteostasis [37,65,70,71].

## Figures and Tables

**Figure 1 biomedicines-10-00881-f001:**
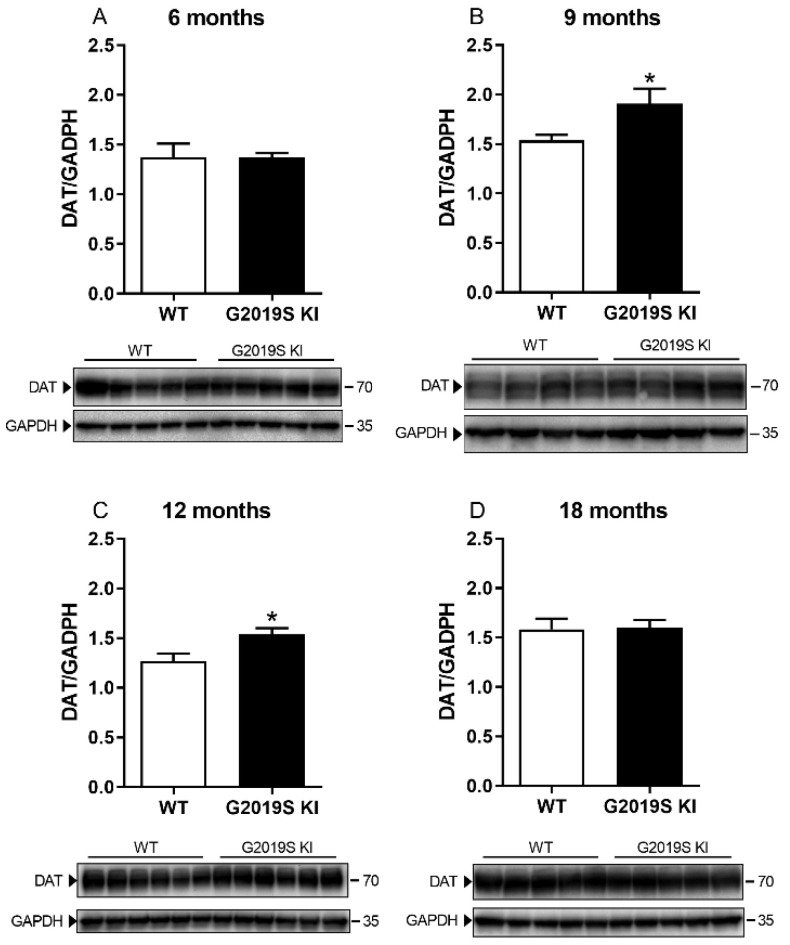
DAT levels are transiently elevated in G2019S KI mice. Western blot analysis of DAT levels in the striatum of 6-month-old (**A**), 9-month-old (**B**), 12-month-old (**C**) and 18-month-old (**D**) G2019S KI mice and age-matched WT controls. Data are expressed as mean ± SEM of *n* = 6 mice per group. Statistical analysis was performed using the Student *t*-test, two-tailed for unpaired data. * *p* < 0.05, different from WT.

**Figure 2 biomedicines-10-00881-f002:**
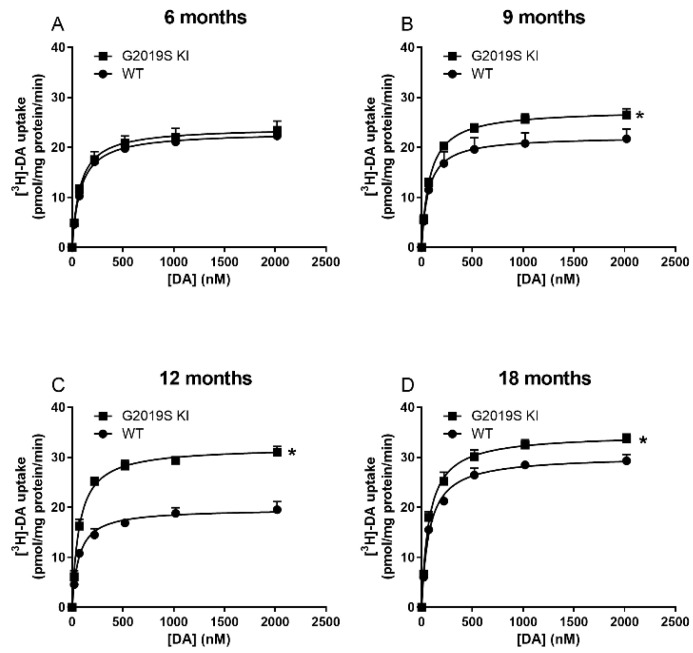
Dysfunctional DAT activity in G2019S KI mice appears at 9 months of age. Kinetic analysis of [^3^H]-DA uptake in synaptosomes were performed in the striata of 6-month-old (**A**), 9-month-old (**B**), 12-month-old (**C**) and 18-month-old (**D**) G2019S KI mice versus age-matched WT controls. Data are expressed as mean ± SEM of *n* = 4 mice per group. Statistical analysis was performed using the Student *t*-test, two-tailed for unpaired data. * *p* < 0.05, different from WT mice.

**Figure 3 biomedicines-10-00881-f003:**
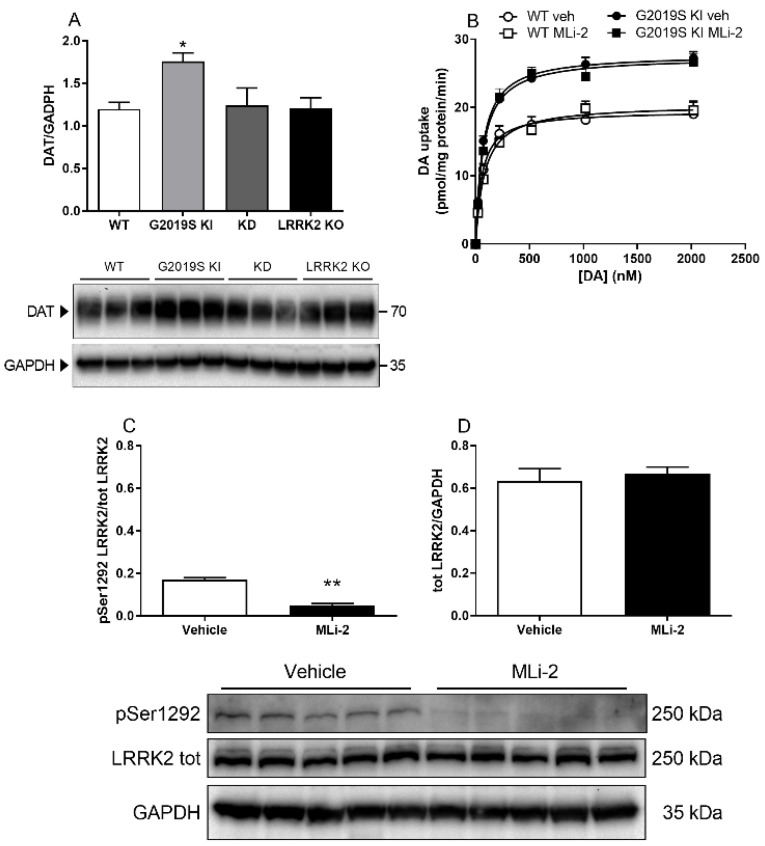
The increase of DAT levels is associated with the G2019S mutation but not sustained by ongoing LRRK2 kinase activity. (**A**) Western blot analysis of DAT levels in the striatum of 12-month-old G2019S KI, LRRK2 KO and KD mice in comparison with WT controls, and representative blots. Data are means ± SEM of *n* = 6 mice per group * *p* < 0.05 different from WT (one-way ANOVA followed by the Tukey test). (**B**) Kinetic analysis of [^3^H]-DA uptake in striatal synaptosomes of 12-month-old G2019S KI and WT mice acutely treated with MLi-2 (10 mg/Kg i.p.) or vehicle. Data are expressed as mean ± SEM of *n* = 4 mice per group. (**C**,**D**) Western blot analysis of pSer1292 LRRK2 and total LRRK2 levels in the striatum of G2019S KI mice administered with MLi-2 (10 mg/Kg, i.p.) or vehicle, and representative blots. Data are expressed as mean ± SEM of *n* = 5 mice per group. ** *p* < 0.01 different from Vehicle (Student *t*-test two-tailed, for unpaired data).

**Figure 4 biomedicines-10-00881-f004:**
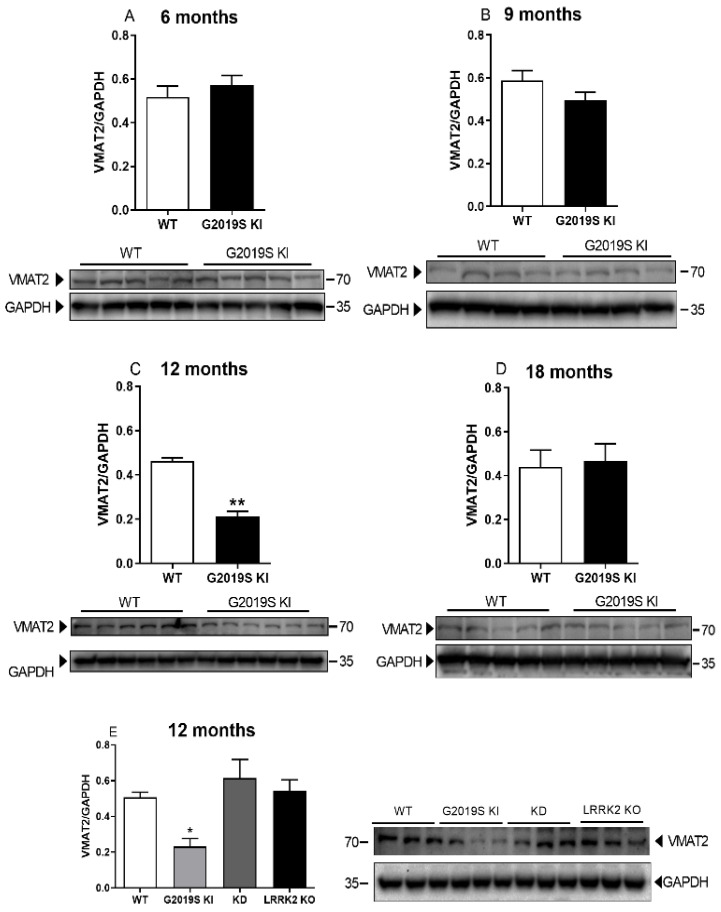
The reduction of VMAT2 levels in G2019S KI mice appears at 12 months and is associated with the G2019S mutation. Western blot analysis of VMAT2 levels in the striatum of 6-month-old (**A**), 9-month-old (**B**), 12-month-old (**C**) and 18-month-old (**D**) G2019S KI mice versus age-matched WT controls, and representative blots. (**E**) Western blot analysis of VMAT2 levels in the striatum of 12-month-old G2019S KI mice versus age-matched LRRK2 KO, KD and WT mice, and representative blots. Data are expressed as mean ± SEM of *n* = 6 animals per group. Statistical analysis was performed using the Student *t*-test, two-tailed for unpaired data (A–D) or one-way ANOVA (**E**) followed by the Tukey test for multiple comparisons. * *p* < 0.05, ** *p* < 0.01 different from WT.

**Figure 5 biomedicines-10-00881-f005:**
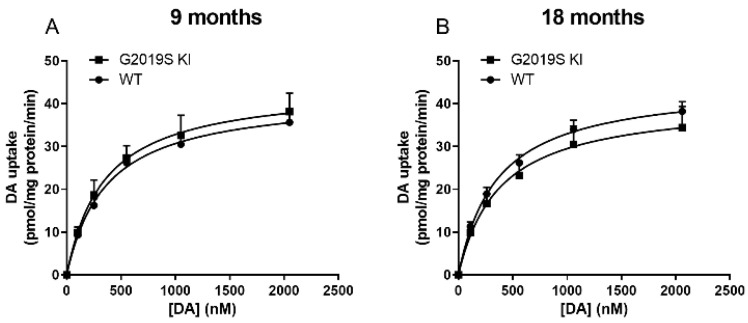
VMAT2 activity is unaltered in 9 and 18-month-old G2019S KI mice. Kinetic analysis of [^3^H]-DA uptake in whole-brain vesicles in the striatum of 9-month-old (**A**) and 18-month-old (**B**) G2019S KI mice versus age-matched WT controls. Data are expressed as mean ± SEM of *n* = 4 animals per group.

**Figure 6 biomedicines-10-00881-f006:**
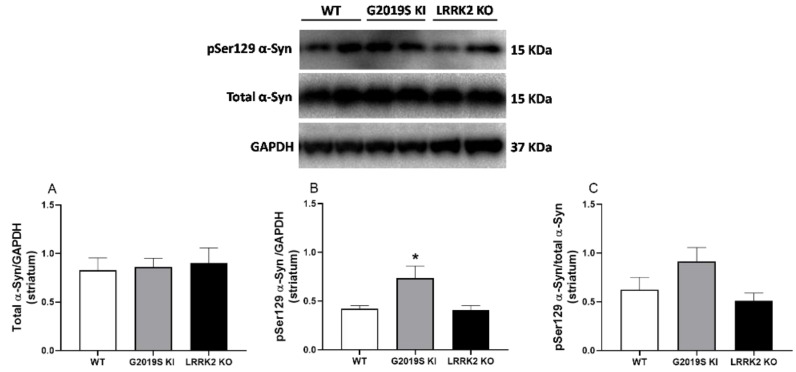
pSer129 α-synuclein levels are elevated in the striatum of 12-month-old G2019S KI mice. Western blot analysis of (**A**) total α-syn and (**B**) pSer129 α-syn levels in the striatum of 12-month-old G2019S KI and LRRK2 KO mice in comparison with age-matched WT controls. (**C**) pSer129 α-syn/total α-syn ratio is also shown. Data are expressed as mean ± SEM of *n* = 8 WT, *n* = 10 G2019S KI and *n* = 8 LRRK2 KO mice. Statistical analysis was performed using one-way ANOVA followed by the Tukey test for multiple comparisons. * *p* < 0.05, different from WT mice.

**Figure 7 biomedicines-10-00881-f007:**
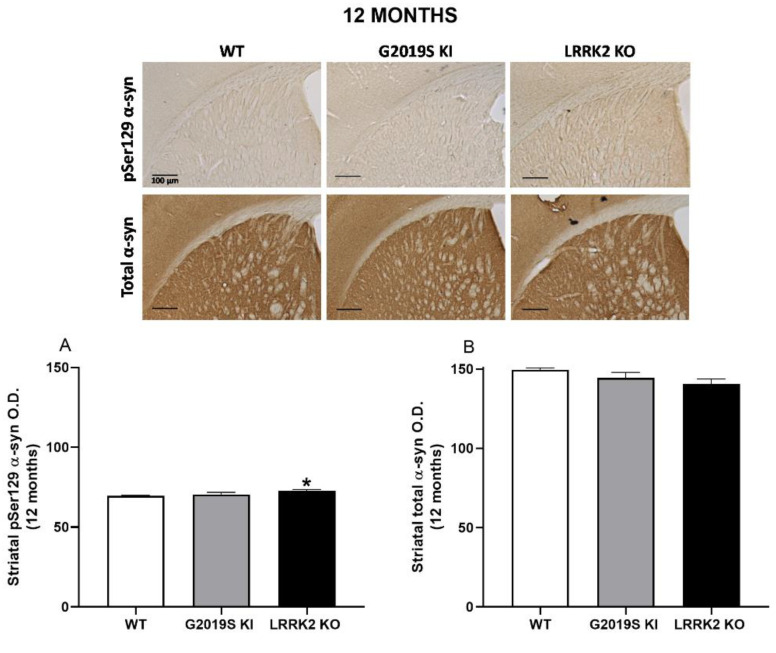
pSer129 α-synuclein levels are elevated in the striatum of 12-month-old LRRK2 KO mice. Immunohistochemistry was performed in striatal slices of 12-month-old G2019S KI and LRRK2 KO mice in comparison with age-matched WT mice. Optical density (O.D.) of pSer129 α-syn (**A**) and total α-syn (**B**) immunostaining was calculated. Data are expressed as mean ± SEM of *n* = 6 mice per group. Representative images were provided in the upper panels (scale bar = 100 µm). Statistical analysis was performed by ANOVA followed by the Tukey test for multiple comparisons. * *p* < 0.05 different from WT mice.

**Figure 8 biomedicines-10-00881-f008:**
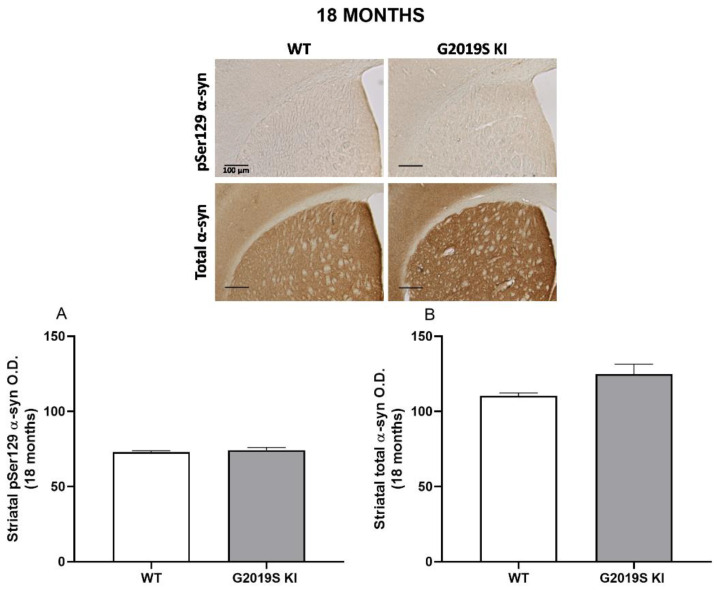
pSer129 and total α-synuclein levels are unchanged in the striatum of 18-month-old G2019S KI mice. Immunohistochemistry in striatal slices of 18-month-old G2019S KI mice in comparison with age-matched WT mice. Optical density (O.D.) of pSer129 α-syn (**A**) and total α-syn (**B**) immunostaining was quantified. Data are expressed as mean ± SEM of *n* = 8 mice per group. Representative images were provided in the upper panels (scale bar = 100 µm).

**Figure 9 biomedicines-10-00881-f009:**
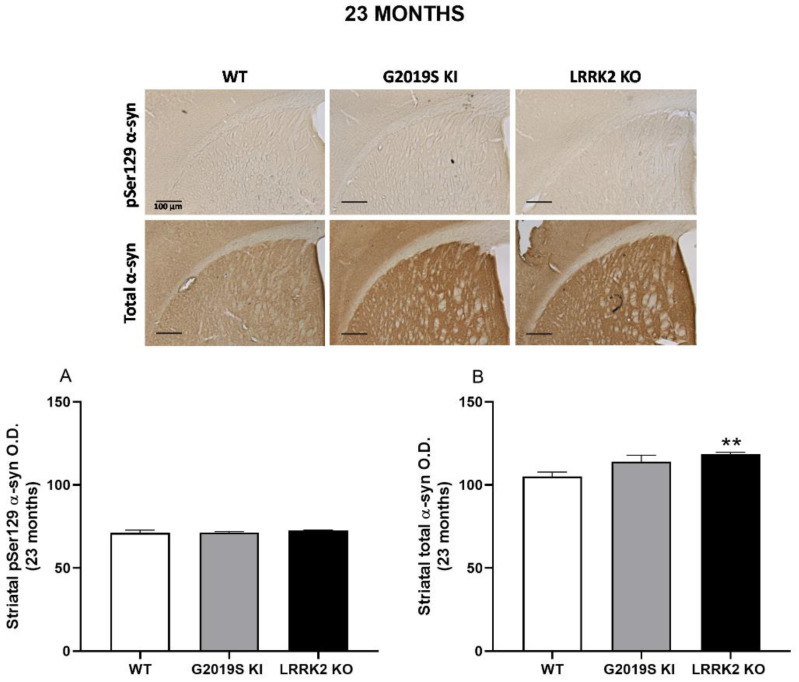
α-synuclein levels are elevated in the striatum of 23-month-old LRRK2 KO mice. Immunohistochemistry in the striatum of 23-month-old G2019S KI and LRRK2 KO mice in comparison with age-matched WT mice. Optical density (O.D.) of pSer129 α-syn (**A**) and total α-syn (**B**) immunostaining was quantified. Data are expressed as mean ± SEM of *n* = 10 WT mice, *n* = 8 G2019S KI mice and *n* = 9 LRRK2 KO mice. Representative images were provided in the upper panels (scale bar =100 µm). Statistical analysis was performed by ANOVA followed by the Tukey test for multiple comparisons. ** *p* < 0.01 different from WT mice.

## Data Availability

The data presented in this study are available on request from the corresponding author.

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
