# Peer review of "Dopamine Transporter, PhosphoSerine129 α-Synuclein and α-Synuclein Levels in Aged LRRK2 G2019S Knock-In and Knock-Out Mice"

_biomedicines, 2022, doi:10.3390/biomedicines10040881_

Round 1

Reviewer 1 Report

The authors present in this revised manuscript the quantification of the levels and activity of the membrane (DAT) and vesicular (VMAT2) dopamine transporters in the striatum of 6-, 9-, 12- and 18-month-old G2019S KI mice; and the levels of pSer129 and total α-synuclein in 12, 18 and 23-month-old G2019S KI mice. Also, they add comparative data with LRRK2 kinase-dead and knock-out mice.

They found DAT dysfunction to emerge at 9 months of age, preceding VMAT2 dysfunction. However, similar results were already presented in a previous paper of the same authors. They also have previously reported that DAT and VMAT2 changes were accompanied by an increase of pSer129 α-syn inclusions. In this manuscript, the authors reported an increase of pSer129 α-syn levels in 12-month-old G2019S KI mice and LRRK2 KO mice; unchanged levels of pSer129 and total α-synuclein levels in 18-month-old G2019S KI mice; and increased levels of total α-synuclein levels in 23-month-old LRRK2 KO mice.

In overall, the paper is well written, with well described methods and results, but I have some concerns, namely:

  • Abstract, lines 28-30 (“Immunoblot analysis showed elevation of pSer129 α-syn …however, was not confirmed by densitometric analysis”). The authors should not take conclusions on the levels of pSer129 α-syn when the densitometric analysis do not corroborate the results.
  • On figure 7A and figure 9B, the difference (statistically significant) on pSer129 and α-syn levels seems minor between G2019S KI and LRKK2 KO mice. What is the difference in terms of %? Can the authors add that information?
  • What are the main novel insights/conclusions of this paper, compared to the previous ones?

Also, some minor typos:

Line 53-54 (“Only in old mice (12-21 months) overexpressing hG2019S under the PDGF 53
promoter a 18-50% neuronal loss in SNc was consistently reported”
). Please revise the punctuation and meaning of the sentence.

Line 361-363 (…but not mice where LRRK2 kinase … increase of LRRK2 kinase activity.”). Please revise the sentence.

Reviewer 2 Report

The authors modified the manuscript according to reviewers’ suggestions. The already high quality of the paper was further improved.

Author Response

This manuscript is a resubmission of an earlier submission. The following is a list of the peer review reports and author responses from that submission.

Round 1

Reviewer 1 Report

In this very interesting article, the authors used LRRK2 G2019S KI mice as a model of Parkinson’s disease due to LRRK2 mutations. They found that the experimental animals developed membrane dopamine transporter (DAT) dysfunction, followed by vesicular dopamine transporter (VMAT2) dysfunction and formation of α-syn phosphorylated at Ser129, indicating a causal link between DAT dysfunction and α-syn metabolism in the G2019S mutation of LRRK2.

The study is well designed and conducted, methodology is more than adequate and the discussion is complete, covering all possible areas of interest.

There are only some typo errors (e.g. familiar Parkinson’s disease in the first line of the abstract)

Reviewer 2 Report

In this study, the authors examined the dynamic changes of DAT, VMAT2, synuclein and phospho-synuclein over time in the striatum of LRRK2 G2019S KI mice. By extending their previous findings, the authors confirmed increased levels of DAT but decreased levels of VMAT2 mainly at around 12 months in the LRRK2 G2019S KI mice compared to WT controls, LRRK2 KO and LRRK2 kinase-dead animals. The authors also provided evidence, indicating alterations of synuclein and phospho-synuclein levels at the same time points. This is a well-controlled study revealing a possible LRRK2 G2019S mutation dependent but LRRK2 kinase activity independent regulation on DAT and VMAT2 levels in vivo. The major limitations are lack of mechanistic explanation on the observed phenotypes and demonstration of their relevance to PD pathogenesis, and providing little new insights compared to their previous study (Longo et al., 2017). In addition, several issues regarding the experiments need further clarification.

(1) Previous studies have demonstrated age-dependent changes of striatal DAT and VMAT2 during normal aging. It is unclear whether the observed differences on DAT and VMAT2 levels between the LRRK2 G2019S KI mice and the controls reflect a pathological process or an alteration of the aging-associated DAT and VMAT2 dynamics in different experimental groups.

(2) The phospho-synuclein staining looks unconvincing at all in most figures. S129 phospho-synuclein is an indication of synuclein pathology, which should be largely absent in the WT controls. The staining signal of S129 phospho-synuclein in figure 6 and 8 are therefore mainly reflecting staining backgrounds. Any quantification based on this is therefore pointless.

(3) In addition to point 2, it seems the authors were trying to normalize the S129 phospho-synuclein to the total synulcein, which is very misleading. In fact, the claimed “increase of phospho-synuclein” at 12 months and “decrease of phospho-synuclein” at 23 months in the LRRK2 G2019S KI mice seem to be largely due to the decrease and increase of total synuclein, respectively. As a pathology marker, the absolute S129 phospho-synuclein level would be a more precise and informative.        

Reviewer 3 Report

In this article “Striatal dopamine transporter dysfunction correlates with α-synuclein inclusions in aging LRRK2 G2019S knock-in mice”, the authors use a genetic model of pre-symptomatic PD, a LRRK2 G2019S knock-in mouse for which they previously described that the introduction of this familial mutated form of LRRK2 induces an alteration of the dopamine transport system (increase of DAT and decrease of VMAT2 expression in mice older than 12 months) and an increase of pS129 a-syn. In this follow up work the authors propose to try to correlate dopamine transporter changes with α-syn inclusions.  In this aim they monitored pser129 a-syn levels in 6, 9, 18 and 23-month-old G2019S KI mice and age-matched wild-type controls.

Changes in DAT were observed by monitoring DAT level by immunoblotting, DA uptake assay and immunohistochemistry. Changes in VMAT2 by immunoblotting and VMAT2 activity. pSer129 α-syn and “total” α-syn levels exclusively by immunohistochemical staining.

The results are interpreted by the Authors as indicating an appearance of the DAT dysfunction at 9 months of age, VMAT2 dysfunction and pSer129 α-syn inclusion formation from 12 months.  The DAT dysfunction is not evoked by wt or kinase dead LRRK2, indicating that it would be specific of the mutation, not reversed by an acute inhibition of the kinase activity, and accompanied by a biphasic pattern of DAT levels (increase only at 12 months). The authors suggest a causal link of these dysfunction with the a-syn metabolism (increase of pSer129  and decrease of a-syn level).

In my opinion the variations of the parameters are very weak, often lacking inter-experimental consistence and in a general manner, are overestimated. The data fail to provide convincing evidences of a correlation between dopamine transporter dysfunction and a-syn metabolism and thus of an hypothetical causal link.  In particular the modifications of a-syn phosphorylation and expression level are not clear and the quantifications are questionable from a methodological point of view. In particular, the authors refer to α-synuclein inclusions but the presented data do not show any a-syn inclusions or aggregates that could be cytologically recognizable. We warmly incite the Authors to take a serious look at the literature regarding experimental α-synuclein inclusions induced by the intracerebral injection of PFFs in mice for instance and revealed with reference anti PS129 syn antibodies: these are real subcellular inclusions that can be clearly evidenced in individual neurons or neurites with a very specific shape. In the present work, it looks like the pS129 syn signal measured is mainly background noise.

Specific points:

Fig.1 quantification of immunoblots of DAT shows a transient increase of DAT level at 9 month. The variations are very weak (not directly detectable by observing blots without band quantification & ratio). At 12 months the difference is essentially brought by the decrease in the wt mice while the absolute level in KI mouse seems to come back to the 6 months level. In any case the time dependent fluctuations are of the same size as those observed in wt mice, rendering difficult to assign them a responsibility for a pathological effect.

Fig. 2 the DA uptake assay indeed shows an increase at 9 months and a 12 months while at 18 months this difference decreases due to a similar but delayed increase in the wt mice.  Again, these data seem to suggest different kinetics of the activity changes between the strains rather than the development of a pathological phenotype, and are not quantitatively consistent with the data shown in figure 1.

Indeed, In figure 3A the DAT/GADPH ratio value at 12 months is not the same as in fig 1.

In fig 3B DAT expression level at 12 and 23 months is evaluated by quantifications of immunohistochemistries performed on striatal sections of mice with different genotypes.  The differences are detected by measuring the area over a threshold value. With this methodology, wt mice at 12 months have 2% of positive area vs more than 60% in the 23 months, suggesting à huge increase of DAT expression  which is not shown by the western blot analysis of Fig. 1 ( at least until 18 months).  It is not clear if the authors have used the same threshold value, but in my opinion IHC quantification by thresholding  of a continuous intensity signal  is a procedure leading  to artifacts.

The same type of consideration can be evoked for fig .6 & 7 concerning a-syn quantification. Images do not show any clear-cut difference, and quantification were made by thresholding, or for pSer129 even as a ratio of thresholded area. The biological meaning of this ratio is not clear: for example, comparing this value for fig. 6 and 7 (12 months vs 18 months) the only difference is an increase from 0.4 to 0,6 for the wt mice, while for the LRRK2 G2019S KI, it remains unchanged. Again, a-syn expression levels increase from 2 to 5 between 12 and 18 months. Unfortunately, total a-syn is stained with an antibody which also recognizes beta-syn (see provider information).

Concerning pSer129 phosphorylation, the authors do not show any western blot quantifications and they  did not demonstrate in any manner the presence of a-syn aggregates. This is easily achievable using a combination of careful cytological analyses and biochemical approaches using brain extracts.  In addition, even if ser129 phosphorylation is a marker of synuclein aggregates, this type of modfication  can also concern non amyloid (non aggregated) forms of the protein (especially in experimental animal models).

In conclusion, even the hypothetical pathological mechanism described by the authors could be interesting,  the data shown here do not support the claims of the Authors, and the many overstatements and methodological errors could be misleading to a non focused/specialist reader, even in the field.

Reviewer 4 Report

In this manuscript, the authors study the levels and activity of the membrane (DAT) and vesicular (VMAT2) dopamine transporters in the striatum of 6-, 9-, 12- and 18-month-old G2019S KI mice; and the levels of pSer129 and total α-synuclein in 12, 18 and 23--month-old G2019S KI mice.  They found that DAT dysfunction emerged at 9 months of age and preceded VMAT2 dysfunction and pSer129 α-syn inclusion formation. The increase in DAT was related to the G2019S mutation that is associated with the increase of LRRK2 kinase activity.

Overall, the paper is well written, with well described methods and results, and deserves consideration for the publication. I only have a minor comment:

  1. The DAT and VAMT2 levels are significantly deregulated at 12 months of age, but this difference disappears at 18 months of age. Moreover, the DAT activity is still altered n 18 months of age mice (G2019S KI mice). Can the author discuss these results in the discussion section?